# Investigating Serum and Tissue Expression Identified a Cytokine/Chemokine Signature as a Highly Effective Melanoma Marker

**DOI:** 10.3390/cancers12123680

**Published:** 2020-12-08

**Authors:** Marco Cesati, Francesca Scatozza, Daniela D’Arcangelo, Gian Carlo Antonini-Cappellini, Stefania Rossi, Claudio Tabolacci, Maurizio Nudo, Enzo Palese, Luigi Lembo, Giovanni Di Lella, Francesco Facchiano, Antonio Facchiano

**Affiliations:** 1Department of Civil Engineering and Computer Science Engineering, University of Rome Tor Vergata, 00133 Rome, Italy; cesati@uniroma2.it; 2Istituto Dermopatico dell’Immacolata, IDI-IRCCS, via Monti di Creta 104, 00167 Rome, Italy; f.scatozza@idi.it (F.S.); d.darcangelo@idi.it (D.D.); giancarlo.antoninic@aslroma2.it (G.C.A.-C.); nudomaurizio@gmail.com (M.N.); e.palese@idi.it (E.P.); l.lembo@idi.it (L.L.); g.dilella@idi.it (G.D.L.); 3Department of Oncology and Molecular Medicine, Istituto Superiore di Sanità, Viale Regina Elena 299, 00161 Rome, Italy; stefania.rossi@iss.it (S.R.); claudiotabolacci@tiscali.it (C.T.)

**Keywords:** melanoma markers, cytokines, machine learning, Support Vector Machine, principal component analysis

## Abstract

**Simple Summary:**

In this study, we investigated the expression of 27 cytokines/chemokines in the serum of 232 individuals (136 melanoma patients vs. 96 controls). It identified several cytokines/chemokines differently expressed in melanoma patients as compared to the healthy controls, as a function of the presence of the melanoma, age, tumor thickness, and gender, indicating different systemic responses to the melanoma presence. We also analyzed the gene expression of the same 27 molecules at the tissue level in 511 individuals (melanoma patients vs. controls). From the gene expression analysis, we identified several cytokines/chemokines showing strongly different expression in melanoma as compared to the controls, and the 4-gene signature “*IL-1Ra*, *IL-7*, *MIP-1a*, and *MIP-1b*” as the best combination to discriminate melanoma samples from the controls, with an extremely high accuracy (AUC = 0.98). These data indicate the molecular mechanisms underlying melanoma setup and the relevant markers potentially useful to help the diagnosis of biopsy samples.

**Abstract:**

The identification of reliable and quantitative melanoma biomarkers may help an early diagnosis and may directly affect melanoma mortality and morbidity. The aim of the present study was to identify effective biomarkers by investigating the expression of 27 cytokines/chemokines in melanoma compared to healthy controls, both in serum and in tissue samples. Serum samples were from 232 patients recruited at the IDI-IRCCS hospital. Expression was quantified by xMAP technology, on 27 cytokines/chemokines, compared to the control sera. RNA expression data of the same 27 molecules were obtained from 511 melanoma- and healthy-tissue samples, from the GENT2 database. Statistical analysis involved a 3-step approach: analysis of the single-molecules by Mann–Whitney analysis; analysis of paired-molecules by Pearson correlation; and profile analysis by the machine learning algorithm Support Vector Machine (SVM). Single-molecule analysis of serum expression identified IL-1b, IL-6, IP-10, PDGF-BB, and RANTES differently expressed in melanoma (*p* < 0.05). Expression of IL-8, GM-CSF, MCP-1, and TNF-α was found to be significantly correlated with Breslow thickness. Eotaxin and MCP-1 were found differentially expressed in male vs. female patients. Tissue expression analysis identified very effective marker/predictor genes, namely, IL-1Ra, IL-7, MIP-1a, and MIP-1b, with individual AUC values of 0.88, 0.86, 0.93, 0.87, respectively. SVM analysis of the tissue expression data identified the combination of these four molecules as the most effective signature to discriminate melanoma patients (AUC = 0.98). Validation, using the GEPIA2 database on an additional 1019 independent samples, fully confirmed these observations. The present study demonstrates, for the first time, that the *IL-1Ra*, *IL-7*, *MIP-1a*, and *MIP-1b* gene signature discriminates melanoma from control tissues with extremely high efficacy. We therefore propose this 4-molecule combination as an effective melanoma marker.

## 1. Introduction

Melanoma is the most aggressive skin cancer with a good prognosis when early diagnosis is achieved. While relevant advances come from newly available therapies, novel approaches are necessary to improve early diagnosis and therapeutic efficacy. Several studies addressed the complex role that specific cytokines and growth factors may play in melanoma biology, acting either as pro- or anti-proliferation and either positively or negatively regulating the immune response [1]. For instance, CXCL10 (IP-10) exerts both pro-and anti-melanoma effects, mostly due to splice variants of its CXCR3 receptors [2]. Several cytokines/chemokines and corresponding receptors are known to be expressed in melanoma tissue, to regulate the multifaceted machinery coordinating the proliferation rate, the angiogenic response, the inflammatory response, the immune response, and the metastatic diffusion [1,3]. Simultaneous quantification of several cytokine/chemokine analytes has recently become available in serum as well as in tissue samples. Previous studies report gene expression profiles identifying low- vs. high-risk patients. For instance, the 31 GEP prognostic classifier identifies BAP1b, MGP, SPP1, CXCL14, CLCA2, S100A8, BTG1, SAP130, ARG1, KRT6B, GJA1, ID2, EIF1B, S100A9, CRABP2, KRT14, ROBO1, RBM23, TACSTD2, DSC1, SPRR1B, TRIM29, AQP3, TYRP1, PPL, LTA4H, and CST6 [4]. Other studies investigated the expression of orphan receptors as well as known chemokine receptors and chemokine ligands in melanoma metastases, leading to the identification of several molecules differentially expressed in metastatic melanoma, such as GPR18, GPR34, GPR119, GPR160, GPR183, P2RY10, CCR5, CXCR4, CXCR6, CCL4, CCL5, CCL14/15, CXCL8, CXCL9, CXCL14, and XCL1/2 [5]. An additional study reports a significant expression change of six chemokines (namely, CCL2, CCL3, CCL4, CCL5, CXCL9, and CXCL10) related to the lymphocyte infiltration in the melanoma tissue [6]. Statistically significant differential plasma expression in melanoma patients vs. controls has been reported for IL-2, IL-6, and IL-10 [7]. Despite the high statistical significance of the differences, none of these molecules show relevant AUC values according to ROC analyses; therefore, to date, they cannot be proposed as markers with clinical relevance.

Additional studies carried out in melanoma patients identified the serum expression level of proinflammatory cytokines, such as IL-2Ra, IL-12-p40, and IFN-α, as good predictors of relapse-free survival [8]. In another study carried out in 40 patients, the serum expression levels of 115 analytes were investigated, including most of the known cytokines and chemokines, such as IL-6, IL-7, IL-10, IL-16, TNF- α trimer, IL-1b, IFN-γ, IL-4R, IL-18, RANK-L, IL-1b, IL-2R, IL-6R, MPIF-1, Leptin, MIG, GDNF, MIP-1 alpha, MIP-1b, MIP-1 delta, ITAC, GM-CSF, MCP-4, MIP-3a, MIP-3b, MMP-1, SP-C, amphiregulin, RANK, MCP-2, IP-10, OPG, FGF-2, and many others. The serum expression profile of TNF-α receptor II, TGF-a, TIMP-1, and C-reactive protein was identified as a profile with prognostic value to predict overall survival in melanoma patients, with an Area Under the Curve (AUC) of 0.89 reduced to 0.72 when the leave-one-out cross-validation technique was applied [9]. An additional study indicates the expression of IL-1Ra, IL-2, and IFNa2 as pro-inflammatory cytokines related to the cytotoxicity associated with anti-CTL4 and anti-PD1 combined therapy [10]. Tissue expression of CCR6 and its ligand CCL20 (MIP-3a) were identified as progression predictors in primary melanoma patients [11]. Prostate-specific membrane antigen (PSMA) was identified by immunohistochemistry analysis as a good marker of metastatic melanoma, in 41 Stage III/IV melanoma human specimens [12]. The ROC analysis measured an AUC = 0.82, i.e., a good performance but not good enough to allow its clinical application as a melanoma marker. Consensus among different studies is often difficult given experimental discrepancies on serum/plasma handling or the antibodies’ sensibility/specificity. Using multiplex immune-based technology may overcome these issues, at least in part, by measuring many different analytes within the same sample.

We have previously investigated melanoma markers by in vitro screening [13], as well as by investigating the ion channels [14,15], autophagy-related molecules [16,17], or molecules related to lipid metabolism [18] in populations composed of hundreds/thousands of controls and patients. In the present study, we investigated the cytokine/chemokine protein expression in the serum of 232 controls/melanoma patients recruited in our hospital, and the gene expression on 511 melanoma tissues selected from the GENT2 database. We report here, for the first time, significant differences related to gender, age, and Breslow thickness in the serum-expression dataset. In the tissue-expression dataset, we report, for the first time, a highly relevant gene marker combination, discriminating healthy controls from melanoma patients with an extremely high accuracy, and reaching an AUC = 0.982, according to the ROC analysis.

## 2. Results

The cytokine/chemokine expression in melanoma patients was analyzed to identify molecules with strong and significant differential expression in patients vs. controls. The cytokine/chemokine protein expression in the serum of 232 patients recruited at the IDI hospital and their RNA expression in tissue biopsies of 511 samples from the GENT2 public database were evaluated. The serum expression and tissue expression of the same 27 human chemokines/cytokines were analyzed as a single-molecule analysis, as a paired-molecule analysis, or as a profile analysis, as reported in the cartoon depicted in Figure 1.

### 2.1. Serum Expression: Single-Molecule Analysis of the Cytokines/Chemokines in Melanoma Patients vs. Controls

The serum dataset included the following information: histopathological diagnosis (96 pathological subjects versus 136 controls), sex (112 male and 120 female), age (median 46.5 years, and mean 48.54 years), Breslow’s depth (minimum value 0 mm, maximum 12 mm, median 0.7 mm, and mean 1.34 mm), and the expression values of the 27 cytokines/chemokines expressed as pg/mL. Table 1 summarizes the information on the serum dataset.

Appendix A report more general data (number of samples for each molecule, minimum value, 25% percentile, median, 75% percentile, maximum, mean, standard deviation, and having passed the normality test (or not)) for all controls and all melanoma patients, respectively.

Table 2 reports the mean values of serum expression of the 27 cytokines/chemokines, the statistical significance of the differences, and the AUC according to the ROC analyses. Five molecules show a significantly (*p* < 0.05) different expression in melanoma vs. the controls, namely, IL-1b, IL-6, IP-10, PDGF-BB, and RANTES. The ROC analyses indicated that none shows a good ability to act as a serum marker of melanoma; in fact, an AUC < 0.70 was found in all cases. Nevertheless, the following Breslow-, age-, and gender-specific characterization indicated many statistically significant differences.

The Breslow thickness-related differences are reported in Table 3 and Table 4. Table 3 reports the mean expression of the 27 cytokines in all melanoma patients as a function of Breslow thickness <1 mm vs. >1mm. Three molecules, namely IL-8, MCP-1, and RANTES, show a statistically significant differential expression. As a further characterization, the correlation of Breslow thickness with serum expression was then investigated in all melanoma patients. Expression of IL-8, GM-CSF, and MCP-1 on one site, and TNF-α on the other, shows a significant negative and positive correlation, respectively (Table 4). Appendix A reports the correlations with Breslow thickness in male melanoma and in female melanoma patients and shows that the cytokines with significant correlations are different in males vs. females.

A further analysis was carried out as a function of age, in all melanoma patients and all controls. The Spearman’s correlation index was computed between the age and the expression value of each cytokine. Seven significant correlations were found in the controls involving IL-7, IL-12(p70), IL-13, IP-10, MIP-1a, MIP-1b, and VEGF. Such correlation were mostly lost in the melanoma patients; in fact, the patients showed only two significant correlations with age, namely, IP-10 and G-CSF (Table 5). A similar finding in male controls vs. male melanoma and in female controls vs. female melanoma is reported in Appendix A, showing a strong reduction in the correlation with age in melanoma samples compared to the controls.

Then, a gender-specific analysis was carried out. Namely, expression levels of the 27 cytokines in male melanoma were compared to female melanoma. Table 6 shows that Eotaxin is significantly increased in male vs. female melanoma, and MCP-1 expression is significantly reduced in male vs. female melanoma, highlighting gender-related differences in cytokine/chemokine serum expression.

Altogether, the results reported in Table 2, Table 3, Table 4, Table 5 and Table 6 indicate strong and significant differential expression of several cytokines/chemokines, as a function of Breslow thickness, age, and gender. Such differences support the known role of these molecules in controlling proliferation, immune response, chemotaxis, and inflammation in melanoma samples, and provide molecular insights into the systemic response to melanoma (see the Discussion section).

### 2.2. Serum Expression: Analysis of Paired Molecules by a Correlation Matrix

A correlation analysis was then carried out. Namely, Spearman’s correlations between the expression values of all pairs of molecules were investigated in control and in melanoma patients. Figure 2 and Figure 3 show the molecule pairs exhibiting the highest correlation coefficients in the control and in melanoma patients, respectively. Specifically, Figure 2A,B shows the heatmap of the intersections having a *p* < 0.05 and Spearman’s rank coefficient >0.60, in the control and melanoma samples, respectively. Figure 3A,B show the molecules pair exhibiting a more severe selection, i.e., a correlation with *p* < 0.05 and a coefficient >0.7. In Figure 2, the 27 molecules were roughly clustered according to their biological functions. A higher number of strong correlations appear in the melanoma samples as compared to the controls, involving either immune/inflammatory molecules, chemokines, and angiogenic factors, indicating that the cytokines/chemokines expression network appears to be more reciprocally correlated in the melanoma samples.

### 2.3. Serum Expression: Profile Analysis by SVM

The serum expression of the 27 chemokines/cytokines was then analyzed as a global profile. The SVM supervised learning algorithm was used, performing a simultaneous analysis of all molecules as predictors of melanoma state. A 10-fold cross-validated, linear-kernel SVM search method was carried out, and the missing values were handled in two alternative ways, as specified in the Methods section. The SVM analysis of the sera data improved the classification efficacy as compared to the single-molecule analysis reported in Table 2, leading to an AUC value = 0.761 and an average accuracy of 0.724 with a *p* = 0.108 (reported in bold and underlined in Table 7). This is slightly higher than 0.7, i.e., the best AUC value obtained by analysis in the single-molecule data (Table 2). Such a result was obtained by removing the missing values and considering as predictors the age and the expression values of the molecules IL-4, IL-8, IL-9, Eotaxin, FGF-2, IFN-γ, IP10, MIP-1a, MIP-1b, PDGF-BB, RANTES, and VEGF.

The SVM analysis on the serum expression data show that the profile analysis may improve the classification efficacy as compared to the single-molecule analysis, but unfortunately not enough to reach clinically relevant values.

### 2.4. Tissues Expression: Single-Molecule Analysis of 27 Cytokines/Chemokines in Melanoma Patients and Controls

Gene expression of the 27 chemokines/cytokines was then evaluated in tissue biopsies of melanoma samples and in control samples. Expression data were derived from the skin cancers section of the GENT2 database. The interface available at the link http://gent2.appex.kr/gent2/ presents data from all skin cancers combined, reporting analyses of Basal Cell Carcinoma (BCC) pooled with Squamous Cell Carcinoma (SCC), Merkel carcinoma primary, Merkel carcinoma metastatic, primary and metastatic melanoma data, for a total of 810 samples. We therefore extracted data referring to normal skin, primary melanoma, and all other melanoma data, excluding all other skin cancers from the analysis. After such a selection, 511 samples were considered, namely, 201 normal skins, 83 primary/primary in-transit melanoma patients, and 227 metastatic melanoma patients. The median gene expression values of the 27 molecules are reported in Table 8 for the three categories. No other stratifications (such as sex, age, or Breslow thickness) were carried out, given the database limitations reported in the Material and Methods section. Differences of the medians in the categories assessed by Mann–Whitney analysis revealed that most molecules show significantly different median values (*p* < 0.05). ANOVA analysis was also carried out on the three groups, indicating similarly strong significant differences for most molecules investigated (see Appendix A).

A ROC analysis was then carried out for every molecule by comparing the control vs. all melanoma, control vs. primary melanoma, control vs. metastatic, and primary melanoma vs. metastatic melanoma samples. The results are reported in Table 9. Four molecules were found to be very good classifiers of the control vs. melanoma samples, namely, IL-1Ra, IL-7, MIP-1a, and MIP-1b, with AUC values of 0.88, 0.86, 0.93, and 0.87, respectively. The corresponding ROC curves for the control vs. all melanoma samples are shown in Figure 4.

These results indicate that analyzing the gene expression of single molecules identifies relevant and significant differences; in this case, the ability to discriminate melanoma from the controls is much higher than the serum-expression data. Nevertheless, such values are below the threshold commonly indicated for potential clinical application. We then carried out the paired-molecule and profile analysis.

### 2.5. Tissue Expression: Analyzing Paired Molecules by a Matrix Correlation

The analysis of the paired-molecule correlations was then carried out, similarly to what we have done for the sera dataset. The correlations between the expression values of all pairs of molecules in the control and melanoma patients were analyzed by computing Spearman’s rank correlation coefficient.

Figure 5 and Figure 6 show the molecule pairs exhibiting high correlation coefficients in the control and melanoma patients. The heatmap in Figure 5 highlights the pairs with significant correlation coefficients of R > 0.6 with *p* < 0.05. Figure 6 shows a more severe selection, i.e., the pairs with a correlation *p*-value < 0.05 and coefficient >0.70. As observed in the serum dataset, this analysis indicates that many strong correlations appear in the melanoma samples as compared to the controls, involving immune/inflammatory molecules, chemokines, and angiogenic factors.

### 2.6. Tissue Expression: Profile Analysis by SVM

As in the case of the serum expression data, the SVM supervised learning algorithm was used to investigate all molecules simultaneously as melanoma predictors. Impressive results were achieved by analyzing the tissue-expression data. The results are shown in Table 10. The average of the AUC values obtained in the 10 iterations of the cross-validation procedure is 0.99, much higher than the highest AUC value obtained in the ROC analysis of the single molecules (namely, 0.93; see Figure 4 and Table 9). The *p*-value is <0.00001; hence, we are highly confident about the statistical significance of this observation.

We then conclude that by using all molecules as melanoma classifiers, the accuracy of the prediction is extremely strong. The ROC curve of the predictive model based on the simultaneous analysis of all molecules is shown in Figure 7.

A model based on many predictors (in this case 27) may present some practical issues. We thus performed a Recursive Feature Elimination (RFE) procedure (summarized in the Methods section), to select the most relevant molecules of the predictive SVM-based model. According to this analysis, the most sensible predictors are, in order, *MIP-1a, IL-1RA, IL-7, MIP-1b, IL-12(p70),* and *TNF-α*. The best four molecules correspond to the ones shown in Table 9, obtained with ROC analyses of the single molecules. As shown in Figure 8, a model based on two molecules, namely, *MIP-1a* and *IL-1RA*, achieves an AUC value = 0.965, while a 4-predictor model *(MIP-1a, IL-1RA, IL-7,* and *MIP-1b*) reaches an AUC = 0.982. These molecules stably represent the best 4-marker combination with the highest AUC value. Further increasing the number of predictors does not add any relevant improvement (see Figure 8).

We therefore conclude that combined analysis of the expression of the *MIP-1a, IL-1RA, IL-7,* and *MIP-1b* genes represents the best combination within the 27 investigated, able to very effectively discriminate the control from the melanoma samples.

We finally investigated the role of the expression of these four genes as a prognostic factor. According to the survival analysis tool available in the GEPIA2 database, 3 out of 4 show significant Hazard Ratios. Namely, *IL7*, *MIP-1a* (*CCL3*), and MIP-1b (*CCL4*) show a HR of 0.71, 0.65, and 0.5, respectively, with *p* = 0.01, 0.002, and 1 × 10^−7^, respectively. These data indicate significantly improved survival in patients with high expression values for these three genes.

### 2.7. Results Validation

The expression of the four molecules reported in Figure 6 was then investigated in an independent database, namely, GEPIA2 (found at http://gepia2.cancer-pku.cn/). Expression was confirmed to be significantly different in melanoma compared to the healthy controls, for *IL-1Ra* (recognized as *ILRn* by GEPIA2), *IL-7*, *MIP-1a* (recognized as *CCL3* by GEPIA2), and *MIP-1b* (recognized as *CCL4* by GEPIA2) (see Figure 9).

The combined expression of these four molecules was then subject to a PCA analysis carried out by the “Dimensionality reduction” tool in GEPIA2. The three most relevant components very effectively differentiated melanoma from controls (Figure 10), indicating that the combined analysis of these four molecules may represent an effective melanoma marker.

This observation fully validated the SVM analysis reported in Figure 6.

## 3. Discussion

While several serum biomarkers are investigated in melanoma patients [19], diagnostic markers currently applied in clinics are restricted to S-100, HMB-45, Melan-A, and SM5-1 [20,21], and the prognostic markers to monitor melanoma progression are S100B, MART1, PMEL, and S100A13 [22]. As recently reported [23], potential markers in melanoma are mutations (on BRAF, NRAS, KIT, GNA11/GNAQ, NF1, CDKN2AI, immunohistochemical biomarkers (such as PD-11 and PD-L1, as well as mutated BRAF and NY-ESO-1), miRNAs, and other serum molecules. The key role cytokines/chemokines play in the immune/inflammatory response and in proliferation and chemotaxis control has been largely investigated; nevertheless, their role as diagnostic or prognostic markers remains to be elucidated. We and others demonstrated the key role of growth factors such as FGF-2, PDGF, and TNF-α in controlling melanoma growth [24,25,26] and melanoma aggressiveness [27]. The present study is the first, to our knowledge, presenting a signature of four cytokines/chemokines as an extremely effective melanoma marker, in a large patient collection. The present study measures cytokine/chemokine expression in serum and in tumor biopsies. We did not expect that the same cytokines/chemokines would be modified in serum and in tissues. In fact, the molecules measured in the serum are likely produced as a systemic response, while the molecules measured within the biopsies are directly produced in the tumor or in the regions immediately close to it. Therefore, the cytokines/chemokines measured within the biopsies reflect more directly the tumor biology and its aggressive behavior. On the contrary, the cytokines/chemokines measured in the serum reflect more how the organism responds to the tumor from an inflammatory/immunological point of view. We cannot exclude that molecules produced within the primary tumor may reach the blood. However, such signals may be not measured due to the large dilution in the blood stream and their expression values may fall below the detection limit. We used the xMAP technology for quantification in serum samples, to minimize as much as possible the sensitivity limitations.

For the sake of clarity, we will discuss below the results of serum expression separately from the results on tissue expression.

Serum expression: Analyzing the serum expression of 27 cytokines/chemokines did not identify any relevant marker when individually analyzed. Nevertheless, significant and strong differential expressions were found in melanoma vs. controls (see Table 2), as well as in melanoma samples as a function of Breslow thickness (see Table 3 and Table 4) or age (Table 5). Furthermore, significant gender-specific differences were identified in Eotaxin and MCP-1 expression (Table 6), as well as in GM-CSF, TNF-α, IL-9, MIP-1a, IL-8, PDGF-BB, and MCP-1 (Appendix A). This finding indicated the molecular bases possibly underlying the different incidence and different mortality rates in male vs. female melanoma [28,29,30,31,32], as well as the unexpected better response of immunotherapies in men than in women [33].

Serum expression of IL-1b, IL-6, IP-10, PDGF-BB, and RANTES was found to be significantly different in melanoma vs. controls (Table 2), reinforced by a much larger patient cohort, since previous observations were carried out on much smaller patient cohorts [34,35]. These molecules make a proinflammatory milieu previously found in uveal melanoma [36] and such findings agree with recent data showing that serum inflammation markers are strongly associated with melanoma progression [37]. Cytokines and chemokines are closely engaged within a large network where several ligands share few receptors [38], therefore reciprocally modulating their pro-, anti-inflammatory, chemotactic, and angiogenic functions [39,40,41]. The chemokines network is known to mediate melanoma interaction with the surrounding tissues [42]. Investigating cytokine/chemokine serum expression may therefore reveal the coordinated tissue reaction to the presence of melanoma. In the present study, Spearman’s correlation matrix revealed for the first time that strong and significant correlations of the expression values are more numerous in melanoma samples than in healthy controls (Figure 2 and Figure 3). In the melanoma samples, molecules involved in inflammation, chemotaxis, and angiogenesis had strong and significant correlations, namely, according to Figure 3, strong correlations of IL-10 with FGF-2, IL-10 with RANTES, IL-5 with IL-13, IL-6 with TNF-α, and of IFN-γ with MCP-1 appear in melanoma samples. Particularly interesting is the strong correlation of IL-6 to TNF-α in the melanoma samples; these two molecules are known to control the ability to evade the immune system control in a PDL-1-dependent manner [43]. Such correlation data indicated that the cross-talk of cytokines and chemokines is altered in melanoma and this may help in defining a melanoma-specific, correlation-matrix fingerprint. We therefore analyzed the entire panel of 27 cytokines/chemokines with the SVM machine learning algorithm, to investigate simultaneously all molecules as predictors of melanoma state. In other studies, SVM effectively discriminated melanoma on the basis of dermoscopic images [44], ultrasonic and spectrophotometric images [45], BRAF status [46], or dermo-fluorescence spectra [47], with a reported accuracy up to 90%. SVM was previously used for prognostic purposes in melanoma patients [48] but, to our knowledge, the present study is the first applying the SVM analysis to cytokine/chemokine-expression values to discriminate melanoma from controls, both in serum and in tissue, in a large group of controls and patients. The SVM procedure was indeed able to improve the ability to classify the serum samples, from AUC = 0.70 for IL-6 expression (see Table 2) up to AUC = 0.761 for the combined indicators (see Table 7). However, this is still not good enough to propose a clinical diagnostic application. We then concluded that the serum expression data of these molecules, while showing strong and significant differences, may not be good classifiers. Several reasons may underlie this result, such as biological reasons (namely, the large serum dilution) as well as technical reasons (namely, samples storage or antibody cross-specificity). We cannot exclude that the cytokine serum expression will give improved information with an improved technology. Protein quantification is a rapidly evolving technology with the continuous upgrading of antibody combinations, sensitivity improvement, and protocol optimization. As an example, a 2007 report [49] identified several cytokines differently expressed in melanoma serum using multiplex xMAP technology, in 179 melanoma patients and 378 healthy controls. However, those data merit to be re-evaluated in the light of the currently available multiplex xMAP technology and new antibodies. As an alternative, quantitative proteomics approaches, based on mass spectrometry, indicated proteins differently expressed in melanoma compared to the controls [50]. However, the sensitivity of the latter techniques limits their application for cytokine/chemokine quantification, as compared to immunometric methods. Analyzing serum from melanoma patients aimed at identifying markers suitable for the early diagnosis, using a minimally invasive technique, expressions significantly different were indeed identified. However, we could not identify good markers within the 27 molecules investigated. This may depend on the molecules chosen (i.e., we should probably change targets and focus on other molecules), or it may depend on the high dilution factor in the serum samples.

We should address briefly the age-matching issue. As reported in Table 1, the mean age in the healthy groups and melanoma groups is different. Such difference reflects what the reality is, i.e., cancer patients are generally older than healthy controls, since increased age is a specific cancer risk factor. In the present study, individuals were sequentially enrolled, and controls were individuals with a suspect lesion removed and diagnosed by the pathologist as a not-cancer lesion. To have a similar age distribution in patients and in the control groups, one would be forced to remove several young healthy controls from the dataset (to match the rarely present young melanoma patients) and to remove several old melanoma patients (to match the rarely present old healthy controls). Age-matching would, therefore, strongly decrease the number of individuals analyzed, and alter the actual patient and control age distribution. The SVM analysis reported in Table 7 was also carried out on the age-matched groups, and the results are similar to the ones obtained from unmatched groups (See Appendix A). Furthermore, the matrix analysis reported in Figure 3 was carried out also on the age-matched groups, and the results are identical to the ones obtained from the unmatched groups. We then conclude that age-matching, required for correct statistical analysis, would strongly reduce the group numerosity; it also would abolish a specific risk factor. In addition, the results achieved on the age-matched groups appear very similar or identical to those obtained on the age-unmatched groups, under our experimental conditions.

Tissue expression: Analyzing the tissue expression was much more effective to discriminate melanoma from controls. This is likely related to biological reasons (i.e., the direct analysis of the melanoma bearing tissue), as well as technical reasons (i.e., using a more stable quantification technique). Very high AUC values were calculated, up to 0.92 (see Figure 4), with the single-molecule analysis. Most of the investigated molecules showed a significant differential expression. This finding indicated that, at the tissue level, most of the cytokines/chemokines investigated are strongly altered in melanoma, prompting us to look for a further improvement of their classifier ability. Analysis of paired molecules reinforced what was observed in the sera dataset, showing that high-correlation pairs appear in melanoma while they are almost absent in the controls (Figure 5 and Figure 6). Particularly interesting are the strong correlations involving the chemokines RANTES, MIP-1a, and MIP-1b (Figure 5 and Figure 6).

Then, an SVM analysis on all molecules simultaneously analyzed was carried out. This analysis strongly improved the classification ability as compared to the single molecules. AUC reached an extremely high value of 0.991 when all 27 cytokines/chemokines were simultaneously considered as melanoma predictors. Use of the Recursive Feature Elimination (RFE) [51] procedure allowed us to identify the four best-performing molecules. The combination of IL-1Ra/IL-7/MIP-1a/MIP-1b shows the relevant AUC = 0.982. Interestingly to notice, the expression of IL-1Ra and IL-7 (known anti-inflammatory cytokines) is significantly reduced in the melanoma samples, while expression of MIP-1a and MIP-1b (known inflammatory chemokines) is significantly increased. Previous data demonstrate that CCL3 (MIP-1α) and CCL4 (MIP-1b) control the infiltration of the immune cells by recruiting antigen-presenting cells, including dendritic cells (DCs), to the tumor site via IFN-γ [52]. However, the specific signature made of the two anti-inflammatory and two pro-inflammatory molecules is a novel finding to our knowledge.

Full validation of these results was achieved on an independent dataset, the GEPIA2 database, reporting expression data from 1019 control and melanoma samples. The four molecules IL-1Ra, IL-7, MIP-1a, and MIP-1b were confirmed to have a significant (*p* < 0.0001) differential expression in melanoma (Figure 9), and their combined analysis with the PCA methodology (a different methodology compared to SVM) was found to effectively discriminate the controls from the melanoma samples (Figure 10).

The identification of the relevant gene markers from the tissue-expression data by using a quantitative technique may help improve histological diagnoses. Identification of a 4-gene signature may be a relevant help for pathologists. Measuring expression of these genes represents a quantitative approach that is operator-independent and may be part of an automatic process useful to identify suspect samples.

## 4. Materials and Methods

### 4.1. Patients Selection and Recruitment

Melanoma patients were consecutively recruited at the hospital sections of IDI-IRCCS, according to the procedure approved by the IDI Ethics Committee (CE 287/1 approved 7/04/2009) based on a suspect skin lesion. All patients gave written informed consent. Patients under any pharmacological melanoma therapy were excluded. Serum was collected before the biopsy procedure, aliquoted, and stored at −80 °C. According to the histological analysis, patients were then assigned to the control arm or to the melanoma arm. A total of 232 patients were recruited in the present study.

### 4.2. Serum Handling

A total of 7 mL of peripheral blood were collected from patients. Blood was collected in tubes with no additives of any type. Tubes were taken at room temperature for 2 to 3 h; they were then centrifuged at 15,000 rpm for 15 min; clear yellow color serum was stored in 100 μL aliquots and stored at −80 °C. The red color was considered a hemolysis sign and such sera were then not analyzed in the current study.

### 4.3. Cytokines Quantification in Sera Samples

Sera were obtained from melanoma patients (*n* = 96) and from patients with non-melanoma suspect lesions (*n* = 136) and were analyzed using xMAP technology on the Luminex platform (X200 Instrumentation equipped with a magnetic washer workstation and software Manager 6.1), which allows the simultaneous quantification of many molecules. The commercial kit used was Bio-Plex Pro human cytokine 27-plex panel (Bio-Rad Laboratories, Hercules, CA, USA), able to measure the following analytes: IL-1Ra, IL-1b, IL-2, IL-4, IL-5, IL-6, IL-7, IL-8, IL-9, IL-10, IL-12(p70), IL-13, IL-15, IL-17, TNF-α, IFN-γ, Eotaxin, Macrophage Inflammatory Protein *(MIP)-1a* (*MIP-1a*; *CCL3*), Macrophage Inflammatory Protein *(MIP)-1b (MIP-1b; CCL4),* Monocyte Chemoattractant Protein *(MCP)-1 (CCL2)*, Granulocyte Colony stimulating factor (G-CSF), GM-CSF, Basic Fibroblast growth factor (FGF-2), Interferon γ-induced protein 10 (IP-10; CXCL10), Regulated on Activation, Normal T cell Expressed and Secreted (RANTES), Platelet-Derived Growth Factor (PDGF-BB), and Vascular Endothelial Growth Factor (VEGF). Samples were handled according to the manufacturer’s instructions and as previously reported [27].

### 4.4. Serum Expression Data

The serum dataset was composed of 232 records corresponding to 232 different individuals. The recorded data included in each case the histopathological diagnosis, sex, age, Breslow’s thickness, and the expression values of the 27 chemokines/cytokines, reported in pg/mL. The expression of a few molecules was undetectable in some patients. Specifically, expressions of IL-2, IL-5, IL-6, IL-10, IL-13, and IL-15 were undetected in most cases and were measured in less than 30 controls and/or less than 30 melanoma samples. We handled missing values in two alternative ways, i.e., we removed the missing data point from the analysis (either the whole cytokine from all samples or the whole sample containing the missing value, depending on the performed analysis while trying to maximize the size of the resulting dataset), or, alternatively, we assumed the missing values are equal to zero, thus assuming that all missing values were caused by expression values lower than the measurement thresholds of the diagnostic kits.

### 4.5. Tissue Expression Data from GENT2 Database

The 27 molecules measured in the sera were investigated in control and melanoma samples taken from the GENT2 database, according to transcriptomic data; data of 201 normal skin biopsies were from 11 independent studies (referred to as GSE39612, GSE30355, GSE14905, GSE13355, GSE7553, GSE42109, GSE16161, GSE15605, GSE7307, GSE46239, and GSE7307); data of 83 primary melanoma biopsies were from 4 independent studies (referred to as GSE10282, GSE15605, GSE7553, and GSE62837); data of 227 metastatic melanoma biopsies were from 11 independent studies (referred to as GSE62837, GSE7307, GSE31879, GSE38312, GSE15605, GSE35640, GSE7553, GSE4587, GSE19293, GSE19234, and GSE22968). The tissue dataset was composed of 511 records, each describing a single subject: the recorded data included the histopathological diagnosis, sex, age, melanoma stage, and the expression values of the 27 cytokines. Each record always states whether the subject has been clinically diagnosed as affected by melanoma (310 patients) or not (201 controls). In this dataset, there were no missing values within the expression data. However, sex and age data were not recorded in most cases: only 217 records reported sex (138 males and 79 females) and only 125 records indicated the age (minimum 20 years, maximum 92 years, median 56 years, mean 59.56 years), and the majority of the control subjects had no sex and age data (about 70% of total). Therefore, sex and age stratifications were not carried out in the tissue-expression data.

### 4.6. Statistical Analyses

All statistical analyses were carried out on the “R” package version 4.0.0 [53].

#### 4.6.1. Single-Molecule Analysis

Expression data of the single molecules were analyzed as an evaluation of the statistical significance of the median differences, by the Mann–Whitney test with Bonferroni correction. ANOVA analysis of the differences between the controls, primary melanoma, and metastatic melanoma samples in the tissue data was also performed and reported in Appendix A. In this case, normal distribution was evaluated by the Shapiro–Wilk test, and homoscedasticity (homogeneity of variance) was evaluated by the nonparametric Levene test. When the ANOVA analysis showed a significant difference between the medians, Mann–Whitney with Bonferroni correction and Tukey’s honest significance tests were applied as post-hoc tests.

#### 4.6.2. Paired-Molecule Analysis

A data-matrix analysis investigated whether the expression correlations of all molecule pairs show any relevant difference between the control and melanoma samples. Spearman’s rank correlation coefficient was calculated.

#### 4.6.3. Profile Analysis

Profile analysis was based on the Support Vector Machine (SVM) supervised learning algorithm, using a linear kernel [54,55]. Briefly, the method finds the best separation hyperplane between the set of control samples and the set of melanoma samples. Each sample is assumed as a single point in the hyperspace of dimensions n, where n is the number of features that can be used as predictors (specifically, the expression values of the molecules, and optionally age and sex). The result of the SVM algorithm is a separation hyperplane that maximizes the cumulative quadratic distance between the boundary points and the hyperplane itself. A parameter C plays a crucial role when the points are not linearly separable: C represents the tradeoff between decreasing the quadratic distance and ensuring that the boundary points are properly classified. We tuned the parameter C by testing 40 values between approximately 10^−14^ and 10^5^ and selecting the value yielding the largest Area Under Curve (AUC) of the Receiver Operating Characteristic (ROC).

The missing expression values were removed from the dataset, according to two alternative approaches. In the first approach, we removed either the entire sample or, if more convenient in terms of resulting dataset size, all expression values of a specific molecule from the dataset. In the second approach, we assigned all missing values to zero [56]. The expression values of each molecule were then transformed to have average = 0 and standard deviation = 1, according to standard methods for this kind of analysis [57,58]. To validate the results, a 10-fold cross-validation procedure was applied. The SVM algorithm considers all predictors as coordinates in a multidimensional space, hence the prediction model is based on the whole set of 27 expression values of the molecules. For the analysis of the tissue-expression data, a Recursive Feature Elimination procedure [51] was applied to identify the molecules having the greatest impact. Therefore, the most relevant molecules of the predictive SVM-based model were identified. This method essentially repeats several times the cross-validated SVM analysis by excluding one of the predictors at a time, then discarding the weakest one, and restarts the whole process on the set of remaining predictors. By this procedure, the molecules having the weakest impact on the performance of the SVM model were identified and removed from the feature set.

### 4.7. Results Validation

#### 4.7.1. Cross-Validation Procedure

All statistical results involving random selections of samples (namely SVM analyses) were validated using “cross-validation” methods to reduce errors due to overfitting. Overfitting errors are caused by an over-optimization of the parameters of a statistical method that achieves an optimal result on the available dataset but poor results on a dataset built from a different set of observations. In a typical k-fold cross-validation procedure, the dataset is randomly partitioned in k subsets of approximately equal size. The statistical method is then repeated k times: in every execution, k-1 subsets are used as “training sets” to optimize the parameters of the method, while the remaining “testing”’ subset is used to evaluate the performances of the method. Every execution of the method uses a different subset as the “testing” set. Eventually, the performances of the method are taken as the average performances of all k executions. In this work we selected k = 10; thus, every measure is the net results of 10 experimental runs.

#### 4.7.2. Validation

The tissue-expression results obtained from the data collected from the GENT2 database were validated on an independent database, GEPIA2 (available at http://gepia2.cancer-pku.cn/#index), reporting the RNA expression data from 461 controls and from 558 melanoma patients. Expression and dimensionality reduction by PCA analysis were carried out by the specific tools available at the GEPIA2 database.

## 5. Conclusions

We report here, for the first time, significant differences in cytokine expression as a function of the pathological state and gender, or age, or Breslow thickness, in the serum expression of a large patient dataset. Such differences are likely related to the systemic response to the tumor and may help, at least in part, investigating the known heterogeneity of this tumor. Furthermore, by analyzing gene expression in a large tissue expression dataset, we report, for the first time, a highly relevant 4-gene signature that discriminates the controls from the melanoma patients. We also show here that the machine learning algorithm SVM appears to be very effective in improving the classification ability for potentially diagnostic purposes and clinical applications.

## Figures and Tables

**Figure 1 cancers-12-03680-f001:**
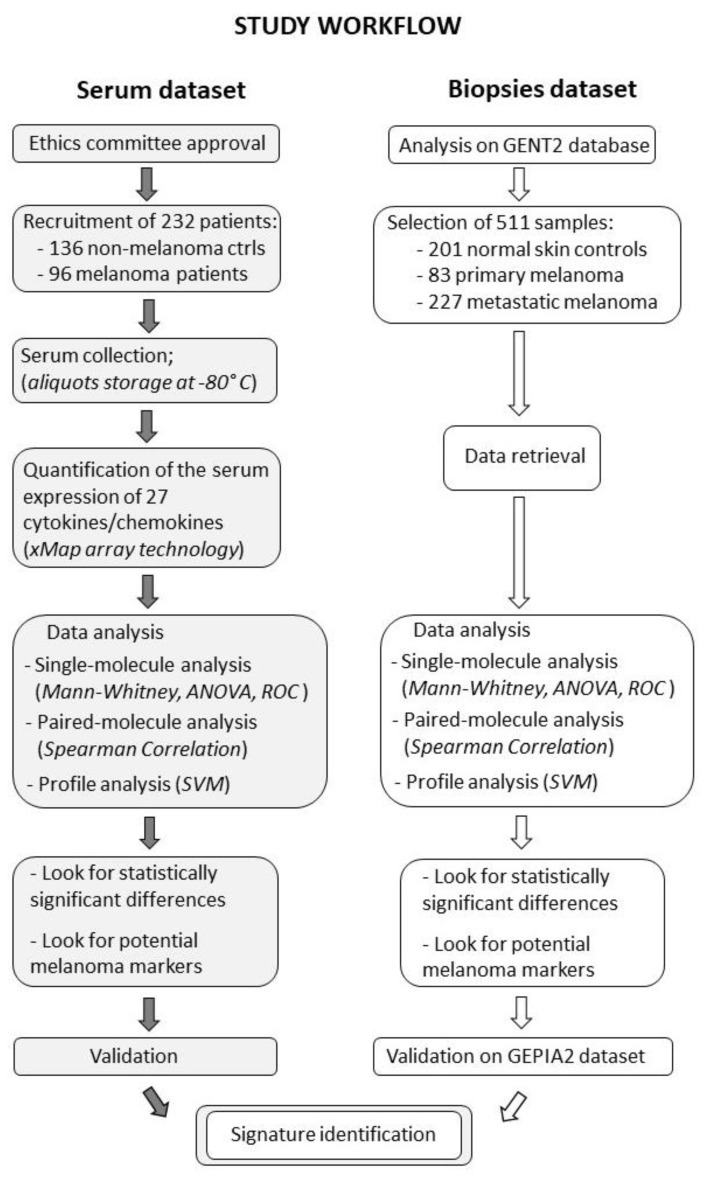
The cartoon reports the study workflow for the serum and tissue samples.

**Figure 2 cancers-12-03680-f002:**
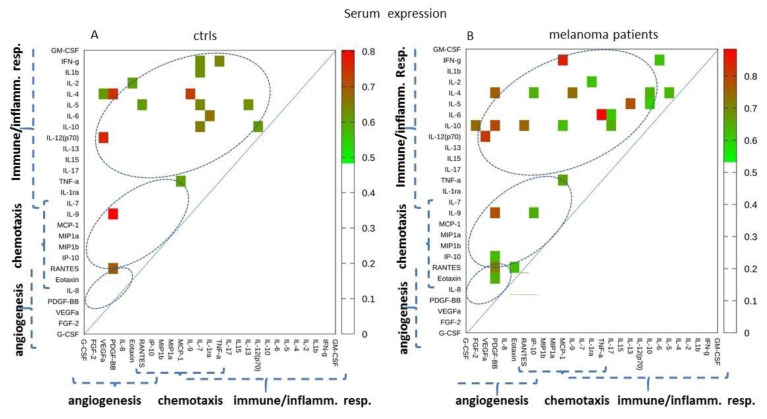
Correlation analysis of the serum expression values. The heatmaps in (**A**,**B**) show the correlation of pairs of expression values having a *p*-value <0.05 and Spearman’s rank coefficient >0.60, in the control and in melanoma samples, respectively.

**Figure 3 cancers-12-03680-f003:**
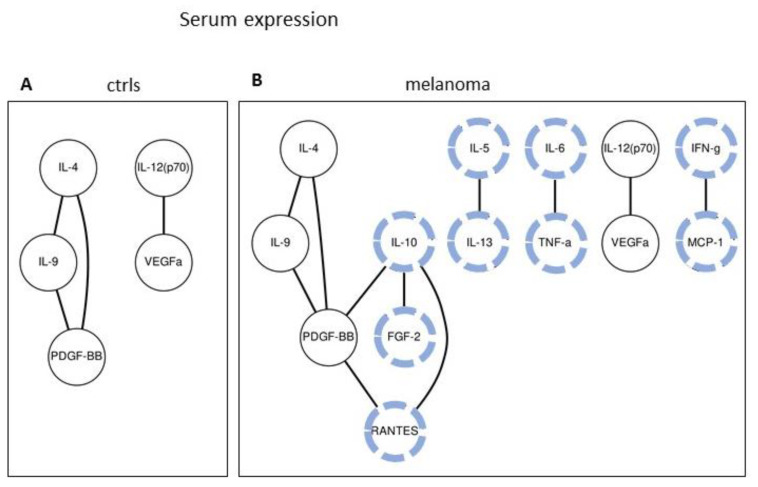
Correlation analysis of the serum expression values of all the control (**A**) and melanoma (**B**) samples. The connected circles show the paired molecules having *p* < 0.05 and Spearman’s R coefficient >0.70, in the control and melanoma samples, respectively. The correlations specifically present in the melanoma samples are highlighted with blue dashed lines.

**Figure 4 cancers-12-03680-f004:**
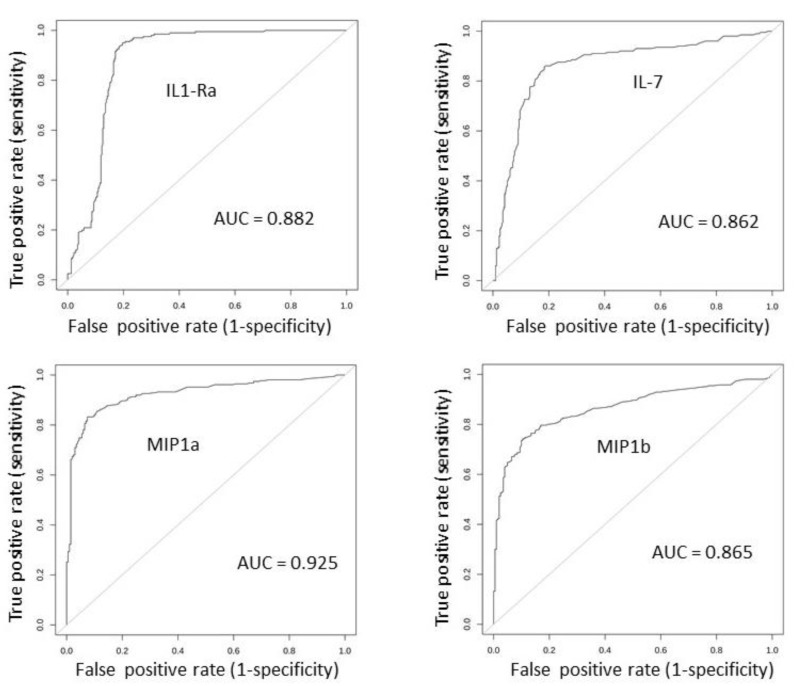
AUC plots of the IL-1Ra, IL-7, MIP-1a, and MIP-1b gene expression in the controls vs. all melanoma samples, for the tissue-expression values.

**Figure 5 cancers-12-03680-f005:**
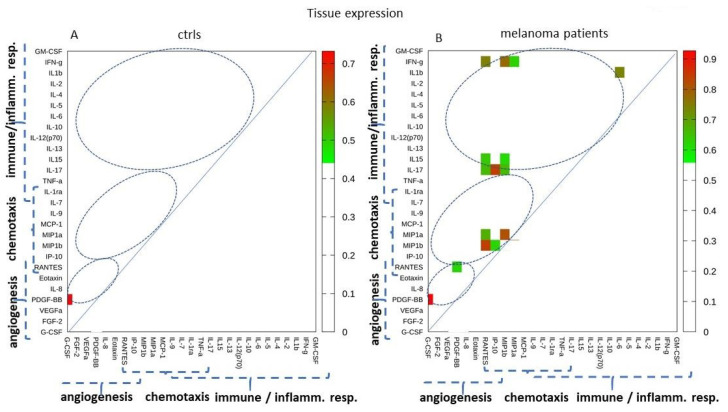
Correlations of the tissue expression values. The heatmap shows the correlation among pairs of expression values with a *p* < 0.05 and Spearman’s rank coefficient >0.60.

**Figure 6 cancers-12-03680-f006:**
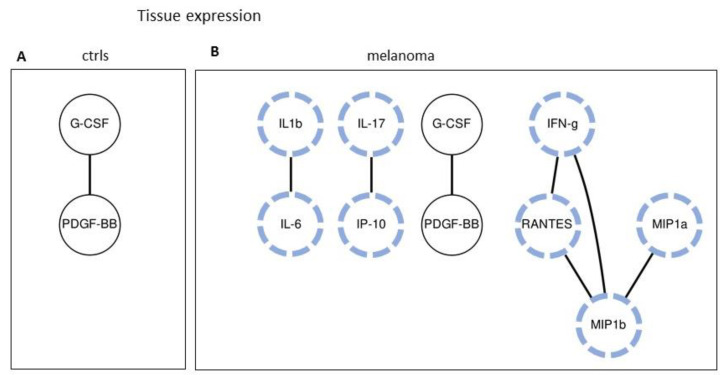
Correlation analysis of the tissue expression values of all the control (**A**) and melanoma (**B**) samples. The connected circles show the paired molecules with a *p* < 0.05 and Spearman’s R coefficient >0.70, in the control and melanoma samples, respectively. The correlations specifically present in the melanoma samples are highlighted with blue dashed lines.

**Figure 7 cancers-12-03680-f007:**
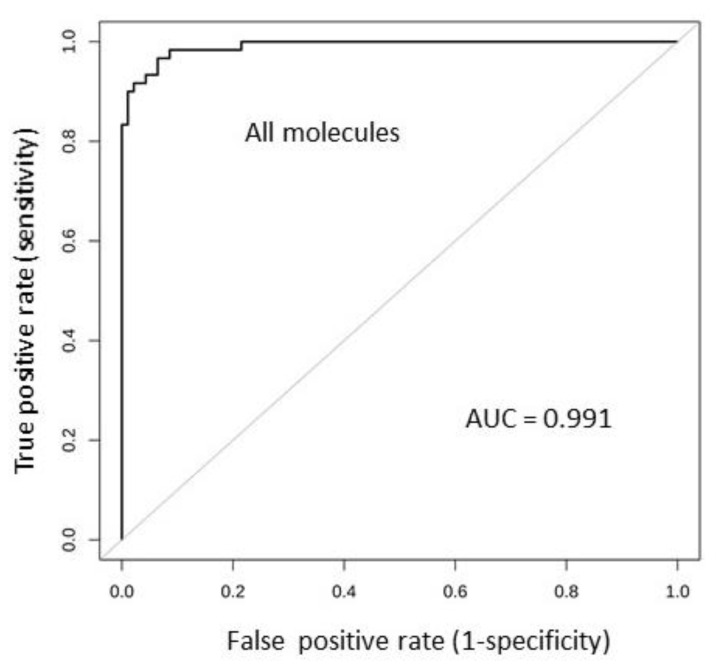
ROC analysis of the SVM-based model, classifying the melanoma samples against the controls.

**Figure 8 cancers-12-03680-f008:**
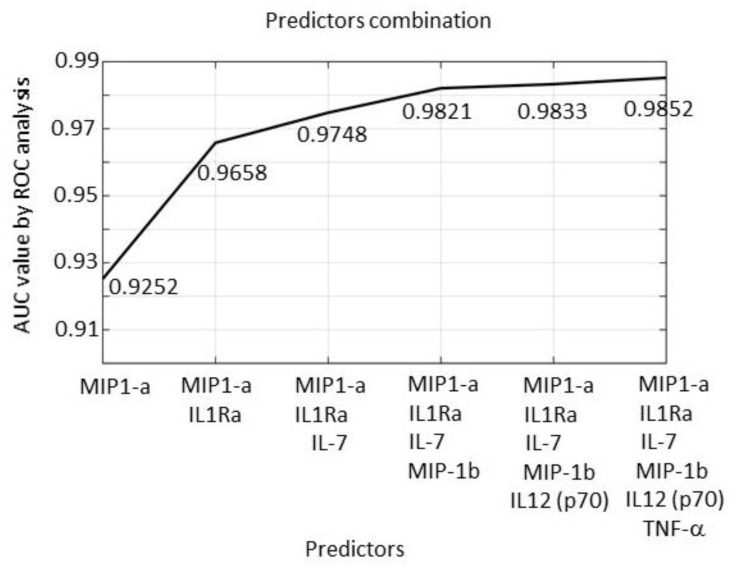
AUC values obtained by the SVM algorithm using one predictor (*MIP-1a*), two predictors (*MIP-1a* + *IL-1RA*), three predictors (*MIP-1a* + *IL-1RA* + *IL-7*), four predictors (*MIP-1a* + *IL-1RA* + *IL-7* + *MIP-1b*), etc., up to six predictors.

**Figure 9 cancers-12-03680-f009:**
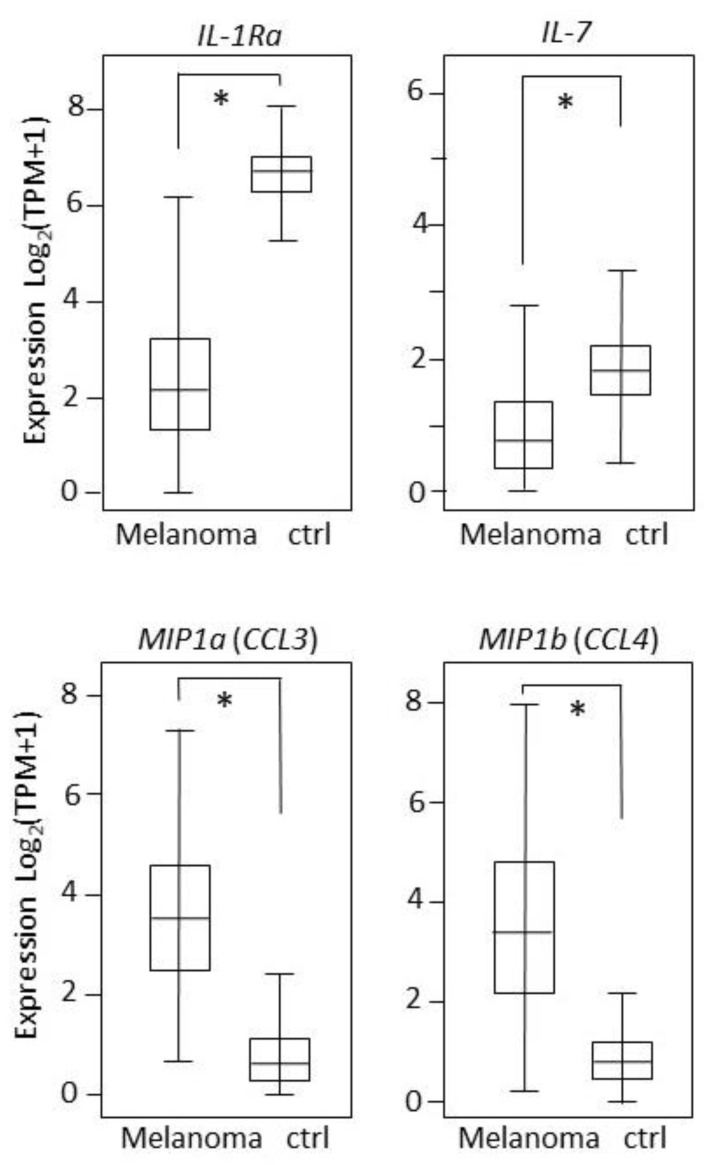
Tissue expression according to the GEPIA2 database. An asterisk (*) indicates *p* < 0.0001.

**Figure 10 cancers-12-03680-f010:**
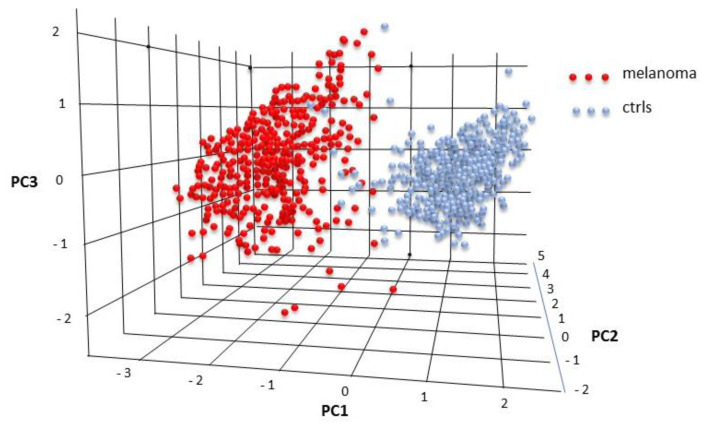
PCA analysis on the tissue expression values of *IL-1Ra, IL-7, MIP-1a,* and *MIP-1b*.

**Table 1 cancers-12-03680-t001:** Descriptive statistics of the population for the serum-expression analysis.

Patient Type	Number	Mean Age	Mean Thickness (mm)	Thickness Distribution	
				<1 mm * Number	≥1 mm * Number
Female controls	72	41.3	0.00	0	0
Male controls	64	45.3	0.00	0	0
Female melanoma	48	54.5	1.60	23	22
Male melanoma	48	58.0	1.08	31	14
Total	232				

* The 1 mm limit is consistent with the current threshold used for staging of T1 melanoma patients and allowed the best case distribution. Not all pathological samples report the thickness value.

**Table 2 cancers-12-03680-t002:** Serum expression. Medians of the expression values of the 27 cytokines/chemokines for the control and melanoma patients. The table also reports the significance of the differences according to Mann–Whitney analyses. For *p*-values < 0.05, the null hypothesis that the control and melanoma patients have the same median should be rejected (significant values reported in bold and are underlined); i.e., a *p* < 0.05 indicates a significant difference. Moreover, the table reports the classification performances as the AUC values from the ROC analyses.

Cytokines	Controls Median (*n* = 136)	Melanoma Median (*n* = 96)	*p*-Value Mann–Whitney Controls vs. Melanoma	AUC ± S.E. by ROC Analysis
**IL-1b**	0.53	0.65	**0.04**	0.61 ± 0.05
**IL-1Ra**	26.77	17.83	0.14	0.56 ± 0.04
**IL-2**	3.45	2.14	0.14	0.66 ± 0.10
**IL-4**	2.95	2.88	0.47	0.53 ± 0.04
**IL-5**	2.77	2.34	0.22	0.60 ± 0.08
**IL-6**	5.37	3.17	**0.04**	0.70 ± 0.09
**IL-7**	2.24	2.24	0.72	0.52 ± 0.05
**IL-8**	6.63	6.4	0.55	0.52 ± 0.04
**IL-9**	45.58	42.05	0.29	0.54 ± 0.04
**IL-10**	7.08	4.56	0.10	0.63 ± 0.08
**IL-12(p70)**	15.74	16.07	1.00	0.50 ± 0.05
**IL-13**	2.43	2.99	0.83	0.52 ± 0.09
**IL-15**	52.22	30.11	1.00	0.50 ± 0.27
**IL-17**	16.72	13.1	0.37	0.54 ± 0.04
**Eotaxin**	95.28	106.28	0.38	0.53 ± 0.04
**FGF-2**	32.68	30.01	0.26	0.55 ± 0.04
**G-CSF**	4.73	5.41	0.33	0.55 ± 0.05
**GM-CSF**	10.21	10.63	0.67	0.53 ± 0.06
**IFN-γ**	19.1	23.2	0.48	0.53 ± 0.04
**IP-10 (CXCL10)**	438.69	501.41	**0.04**	0.58 ± 0.04
**MCP-1(MCAF)**	18.59	12.49	0.24	0.58 ± 0.07
**MIP-1a (CCL3)**	1.78	1.74	0.59	0.52 ± 0.04
**MIP-1b (CCL4)**	54.36	56.26	0.43	0.53 ± 0.04
**PDGF-BB**	1603.74	1033.41	**0.01**	0.61 ± 0.05
**RANTES (CCL5)**	11,353.34	8735.27	**0.01**	0.57 ± 0.06
**TNF-α**	16.4	18.06	0.26	0.52 ± 0.06
**VEGF**	59.75	57.98	0.88	0.54 ± 0.06

Bold underlined: highlight the result.

**Table 3 cancers-12-03680-t003:** Serum expression as a function of Breslow thickness. Count indicates the number of patients analyzed. The median expression values of the 27 molecules with a Breslow thickness <1 mm or >1 mm are reported. The significance of the difference according to the Mann–Whitney analyses is also reported: for *p*-values > 0.05, the null hypothesis that the two distributions of cytokine expressions have the same median should be rejected (significant values are reported in bold and are underlined). In simpler words, when the *p*-value is < 0.05, the cytokine expressions in patients with a Breslow thickness <1 mm and expression in patients with a thickness >1 mm have significantly different medians.

Cytokines	Melanoma Breslow Thickness <1 mm		Melanoma Breslow Thickness ≥1 mm		Mann-Whitney <1 mm vs. ≥1 mm
	Count	Median	Count	Median	*p* Value
**IL-1b**	26	0.65	16	0.72	0.66
**IL-1Ra**	44	18.24	27	17.83	0.59
**IL-2**	5	1.9	7	2.38	0.25
**IL-4**	54	2.85	35	3.06	0.81
**IL-5**	15	2.34	9	1.7	0.86
**IL-6**	7	0.83	8	3.34	0.18
**IL-7**	31	2.24	15	2.9	0.34
**IL-8**	47	7.8	30	5.78	**0.01**
**IL-9**	52	42.55	35	41.91	0.64
**IL-10**	11	3.84	10	5.12	0.92
**IL-12(p70)**	40	20.48	20	14.64	0.51
**IL-13**	11	2.43	8	3.2	0.60
**IL-15**	2	13.02	1	42.31	0.67
**IL-17**	38	13.15	30	11.79	0.94
**Eotaxin**	54	108.94	35	93.24	0.71
**FGF-2**	49	29.73	34	31.46	0.77
**G-CSF**	27	5.45	10	4.12	0.30
**GM-CSF**	17	14.06	25	9.78	0.22
**IFN-γ**	51	23.49	31	22.13	0.51
**IP-10 (CXCL10)**	53	491.81	35	574.3	0.88
**MCP-1(MCAF)**	18	10.56	11	24.83	**0.02**
**MIP-1a (CCL3)**	54	1.82	35	1.71	0.56
**MIP-1b (CCL4)**	53	55.51	35	53.52	0.73
**PDGF-BB**	53	1048.16	35	1080.98	0.47
**RANTES (CCL5)**	53	10,341.43	35	7534.18	**0.03**
**TNF-α**	43	16.65	19	22.78	0.06
**VEGF**	51	63.58	35	52.26	0.45

Bold underlined: highlight the result.

**Table 4 cancers-12-03680-t004:** Serum expression in all melanoma patients (male + female): correlation with Breslow thickness. For *p*-values < 0.05 (reported in bold and underlined), the null hypothesis (i.e., the Spearman’s correlation coefficient R is 0) should be rejected; i.e., cytokine distributions with *p*-values < 0.05 are significantly correlated with Breslow thickness.

Cytokines	No. of Pairs	Spearman R Correlation of Serum Expression with Breslow Thickness	*p*-Value (2-Tails)
**IL-1b**	42	0.04	0.80
**IL-1Ra**	71	0.04	0.75
**IL-2**	12	0.01	0.97
**IL-4**	89	−0.02	0.88
**IL-5**	24	0.01	0.96
**IL-6**	15	0.24	0.40
**IL-7**	46	0.28	0.06
**IL-8**	**77**	−0.23	**0.05**
**IL-9**	87	−0.09	0.42
**IL-10**	21	−0.14	0.55
**IL-12(p70)**	60	−0.02	0.86
**IL-13**	19	−0.09	0.72
**IL-15**	3	1.00	0.33
**IL-17**	68	0.05	0.67
**Eotaxin**	89	−0.01	0.96
**FGF-2**	83	0.03	0.81
**G-CSF**	37	0.04	0.83
**GM-CSF**	42	−0.40	**0.01**
**IFN-γ**	82	0.07	0.53
**IP-10 (CXCL10)**	88	0.04	0.72
**MCP-1(MCAF)**	29	0.39	**0.04**
**MIP-1a (CCL3)**	89	−0.09	0.40
**MIP-1b (CCL4)**	88	−0.02	0.86
**PDGF-BB**	88	−0.10	0.35
**RANTES (CCL5)**	88	−0.20	0.06
**TNF-α**	62	0.31	**0.01**
**VEGF**	86	−0.02	0.87

Bold underlined: highlight the result.

**Table 5 cancers-12-03680-t005:** Serum expression in all melanoma patients (male + female): correlation with age. For *p*-values < 0.05 (reported in bold and underlined), the null hypothesis (i.e., the Spearman correlation coefficient R is 0) should be rejected; i.e., cytokine distributions with *p*-values < 0.05 are significantly correlated with age.

Cytokines	All Controls (Male + Female)			All Melanoma (Male + Female)		
	No. of Pairs	Spearman R Correlation of Serum Expression with Age	*p* Value (2 Tails)	No. of Pairs	Spearman R Correlation of Serum Expression with Age	*p*-Value (2 Tails)
**IL-1b**	85	0.14	0.21	44	−0.13	0.40
**IL-1Ra**	121	0.15	0.10	75	−0.08	0.50
**IL-2**	19	−0.21	0.38	13	0.23	0.46
**IL-4**	135	0.08	0.36	95	−0.05	0.66
**IL-5**	30	0.34	0.07	25	−0.20	0.33
**IL-6**	21	0.00	0.99	16	−0.01	0.96
**IL-7**	81	0.35	**0.001**	48	−0.09	0.53
**IL-8**	128	0.12	0.17	83	−0.20	0.07
**IL-9**	135	0.12	0.17	93	0.05	0.61
**IL-10**	39	0.05	0.77	22	−0.09	0.69
**IL-12(p70)**	110	0.30	**0.002**	62	−0.08	0.55
**IL-13**	27	0.39	**0.04**	19	0.21	0.38
**IL-15**	2	-	-	4	-	-
**IL-17**	108	0.12	0.23	73	−0.01	0.93
**Eotaxin**	132	0.13	0.13	95	−0.01	0.97
**FGF-2**	129	0.02	0.85	88	0.01	0.94
**G-CSF**	92	−0.03	0.76	40	−0.38	**0.02**
**GM-CSF**	48	−0.05	0.72	46	−0.10	0.53
**IFN-γ**	136	0.09	0.31	86	−0.12	0.26
**IP-10 (CXCL10)**	136	0.22	**0.01**	94	0.20	**0.05**
**MCP-1(MCAF)**	47	0.11	0.45	30	−0.17	0.38
**MIP-1a (CCL3)**	133	0.23	**0.01**	95	−0.16	0.13
**MIP-1b (CCL4)**	136	0.30	**0.0005**	94	−0.15	0.15
**PDGF-BB**	135	0.02	0.86	94	−0.11	0.29
**RANTES (CCL5)**	136	−0.02	0.84	94	−0.08	0.42
**TNF-α**	120	0.17	0.06	65	−0.14	0.28
**VEGF**	135	0.17	**0.05**	92	0.09	0.41

Bold underlined: highlight the result.

**Table 6 cancers-12-03680-t006:** Serum expression in male melanoma compared to female melanoma. For *p*-values < 0.05 (reported in bold and underlined), the null hypothesis (i.e., the two distributions have the same median) should be rejected. Briefly, the cytokine distributions for the control and melanoma patients with *p*-values < 0.05 have significantly different medians.

Cytokines	Male Melanoma	Female Melanoma	Mann–Whitney
	Median Value	Median Value	*p*-Value
**IL-1b**	0.63	0.785	0.07
**IL-1Ra**	15.17	24.125	0.25
**IL-2**	1.9	3.09	0.14
**IL-4**	3.05	2.75	0.06
**IL-5**	2.34	2.02	0.36
**IL-6**	3.34	3.0	0.73
**IL-7**	2.24	2.31	0.49
**IL-8**	6.57	6.32	0.64
**IL-9**	44.35	38.84	0.14
**IL-10**	3.84	6.31	0.13
**IL-12(p70)**	16.94	14.57	0.98
**IL-13**	2.19	3.23	0.60
**IL-15**	-	30.11	-
**IL-17**	13.1	12.96	0.67
**Eotaxin**	135.15	91.8	**0.002**
**FGF-2**	32.55	29.73	0.25
**G-CSF**	5.09	5.45	0.60
**GM-CSF**	11.71	9.97	0.47
**IFN-γ**	20.83	27.33	0.26
**IP-10 (CXCL10)**	567.19	468.93	0.13
**MCP-1(MCAF)**	10.995	25.6	**0.05**
**MIP-1a (CCL3)**	1.86	1.7	0.49
**MIP-1b (CCL4)**	56.89	51.09	0.30
**PDGF-BB**	1145.64	900.76	0.06
**RANTES (CCL5)**	9536.67	7879.5	0.19
**TNF-α**	17.355	20.11	0.34
**VEGF**	65.345	50.515	0.49

Bold underlined: highlight the result.

**Table 7 cancers-12-03680-t007:** Results of the SVM method applied to the serum expression dataset. Missing values are either removed or assigned as zero. In the first case, some molecules are removed from the predictor values (remaining molecules are listed in the “Molecules” column). Sex and/or age are or are not regarded as predictors. The *p*-value is the probability that the “Accuracy” value is not significantly above the “No Info Rate” value.

Missing Values	Num. Melanoma	Num. Controls	Training Set Size	Testing Set Size	Predictors: Sex or Age	AUC (ROC)	Accuracy	No Info Rate	*p*-Value
Removed *	72	124	138	58	Sex, Age	0.674	0.621	0.64	0.66
					Sex	0.658	0.638	0.64	0.56
					Age	0.761	0.724	0.64	0.11
					None	0.615	0.586	0.64	0.83
Set to 0 **	96	136	164	68	Sex, Age	0.621	0.588	0.59	0.55
					Sex	0.510	0.588	0.59	0.55
					Age	0.704	0.662	0.59	0.13
					None	0.619	0.588	0.59	0.55

* When the missing values were removed, IL-4, IL-8, IL-9, Eotaxin, FGF-2, IFN-γ, IP-10, MIP-1a, MIP-1b, PDGF-BB, RANTES, and VEGF were simultaneously analyzed by the SVM. **** When the missing values were set to 0, all 27 cytokines/chemokines were simultaneously analyzed by SVM.

**Table 8 cancers-12-03680-t008:** Tissue expression: medians of the expression values of the 27 chemokines/cytokines for the control and for melanoma patients (all, primary, and metastatic). The table also reports the significance of the difference according to Mann–Whitney analyses (Wilcoxon two-sample rank-sum test): for *p*-values < 0.05 the null hypothesis (i.e., the corresponding sets of samples have the same median) should be rejected (values in bold and underlined).

Cytokines	Median				*p*-Value (Mann–Whitney)			
	Ctrls (201)	Melanoma			Ctrls vs. all	(with Bonferroni Correction)		
		All (310)	Prim. (83)	Metast. (227)		Ctrls vs. Prim.	Ctrls vs. Metast.	Prim. vs. Metast.
***IL-1b***	**6.66**	7.04	6.73	7.15	**<0.0001**	0.15	**<0.0001**	0.71
***IL-1Ra***	9.38	7.02	7.03	7.01	**<0.0001**	**<0.0001**	**<0.0001**	1.31
***IL-2***	2.58	2.81	2.58	3.00	0.59	0.96	1.00	1.00
***IL-4***	3.46	3.46	3.17	3.58	0.96	0.45	1.00	0.15
***IL-5***	2.81	3.00	3.00	3.00	0.48	1.00	1.00	1.00
***IL-6***	5.61	6.39	5.98	6.64	**<0.0001**	0.06	**<0.0001**	0.06
***IL-7***	7.22	5.49	5.17	5.73	**<0.0001**	**<0.0001**	**<0.0001**	**0.01**
***IL-8***	3.32	3.32	3.17	3.32	0.99	1.00	1.00	1.00
***IL-9***	2.58	2.81	2.58	2.81	0.76	1.00	1.00	0.48
***IL-10***	3.70	4.88	4.75	5.04	**<0.0001**	**<0.0001**	**<0.0001**	0.66
***IL-12(p70)***	4.25	2.32	2.81	2.00	**<0.0001**	**<0.0001**	**<0.0001**	0.12
***IL-13***	5.64	5.52	5.55	5.49	0.96	1.00	1.00	1.00
***IL-15***	6.95	6.83	6.30	6.97	0.44	**0.001**	1.00	**0.01**
***IL-17***	5.67	6.83	6.07	7.37	**<0.0001**	0.08	**<0.0001**	**0.05**
***Eotaxin***	4.75	5.29	4.95	5.39	**<0.0001**	0.21	**<0.0001**	0.06
***FGF-2***	7.11	7.03	6.25	7.24	0.51	**<0.0001**	0.53	**<0.0001**
***G-CSF***	11.83	11.55	12.25	11.37	0.95	**0.001**	0.30	**0.006**
***GM-CSF***	4.81	4.64	4.52	4.64	0.83	1.00	1.00	0.99
***IFN-γ***	4.70	5.58	4.91	5.83	**<0.0001**	**0.03**	**<0.0001**	**0.01**
***IP-10 (CXCL10)***	2.81	3.86	3.17	4.25	**<0.0001**	0.36	**<0.0001**	**0.01**
***MCP-1(MCAF)***	3.32	3.46	3.00	3.58	**0.03**	1.00	**0.03**	**0.01**
***MIP-1a (CCL3)***	5.13	8.11	7.76	8.24	**<0.0001**	**<0.0001**	**<0.0001**	0.07
***MIP-1b (CCL4)***	5.29	7.90	7.35	8.13	**<0.0001**	**<0.0001**	**<0.0001**	0.12
***PDGF-BB***	13.21	13.01	13.21	12.83	**0.02**	0.60	**0.001**	**0.003**
***RANTES (CCL5)***	6.86	8.16	7.81	8.22	**<0.0001**	**<0.0001**	**<0.0001**	1.00
***TNF-α***	6.13	7.54	7.27	7.57	**<0.0001**	**<0.0001**	**<0.0001**	**0.003**
***VEGF***	8.95	8.97	8.67	9.03	0.37	**0.001**	1.00	**0.002**

Bold underlined: highlight the result; Italics: Genes.

**Table 9 cancers-12-03680-t009:** ROC analysis of the tissue expression data. For each molecule, the AUC values for the ROC analysis was calculated for the following sets of expression values: controls vs. all melanoma, controls vs. primary melanoma, controls vs. metastatic melanoma, and primary melanoma vs. metastatic melanoma samples. For every AUC value, the standard error of the measure obtained by the cross-validation procedure is also shown.

Cytokines	AUC ± S.E. of ROC Analysis			
	Ctrls vs. all Melanoma	Ctrls vs. Primary	Ctrls vs. Metastatic	Primary vs. Metastatic
***IL-1b***	0.62 ± 0.03	0.57 ± 0.04	0.63 ± 0.03	0.54 ± 0.04
***IL-1Ra***	**0.88 ± 0.02**	**0.88 ± 0.03**	**0.88 ± 0.02**	0.53 ± 0.04
***IL-2***	0.51 ± 0.03	0.54 ± 0.04	0.51 ± 0.03	0.53 ± 0.04
***IL-4***	0.50 ± 0.03	0.55 ± 0.04	0.52 ± 0.03	0.57 ± 0.04
***IL-5***	0.52 ± 0.03	0.51 ± 0.04	0.52 ± 0.03	0.51 ± 0.04
***IL-6***	0.64 ± 0.03	0.59 ± 0.04	0.66 ± 0.03	0.59 ± 0.04
***IL-7***	**0.86 ± 0.02**	**0.91 ± 0.03**	**0.85 ± 0.02**	0.61 ± 0.04
***IL-8***	0.50 ± 0.03	0.52 ± 0.04	0.51 ± 0.03	0.52 ± 0.04
***IL-9***	0.51 ± 0.03	0.53 ± 0.04	0.52 ± 0.03	0.55 ± 0.04
***IL-10***	0.68 ± 0.03	0.65 ± 0.03	0.69 ± 0.03	0.55 ± 0.04
***IL-12(p70)***	0.78 ± 0.02	0.77 ± 0.03	0.79 ± 0.02	0.58 ± 0.04
***IL-13***	0.50 ± 0.03	0.50 ± 0.04	0.50 ± 0.03	0.50 ± 0.04
***IL-15***	0.52 ± 0.03	0.64 ± 0.01	0.52 ± 0.03	0.61 ± 0.04
***IL-17***	0.61 ± 0.03	0.54 ± 0.04	0.63 ± 0.03	0.59 ± 0.04
***Eotaxin***	0.63 ± 0.03	0.57 ± 0.04	0.65 ± 0.03	0.59 ± 0.04
***FGF-2***	0.52 ± 0.03	0.67 ± 0.04	0.54 ± 0.03	0.66 ± 0.04
***G-CSF***	0.50 ± 0.03	0.63 ± 0.04	0.55 ± 0.03	0.61 ± 0.04
***GM-CSF***	0.51 ± 0.03	0.52 ± 0.04	0.52 ± 0.03	0.54 ± 0.04
***IFN-γ***	0.69 ± 0.03	0.60 ± 0.04	0.72 ± 0.03	0.61 ± 0.04
***IP-10 (CXCL10)***	0.64 ± 0.03	0.56 ± 0.04	0.67 ± 0.03	0.61 ± 0.04
***MCP-1(MCAF)***	0.56 ± 0.03	0.54 ± 0.04	0.59 ± 0.03	0.62 ± 0.04
***MIP-1a (CCL3)***	**0.93 ± 0.01**	**0.91 ± 0.02**	**0.93 ± 0.02**	0.58 ± 0.04
***MIP-1b (CCL4)***	**0.87 ± 0.02**	**0.87 ± 0.02**	**0.86 ± 0.02**	0.58 ± 0.04
***PDGF-BB***	0.56 ± 0.03	0.55 ± 0.04	0.60 ± 0.03	0.62 ± 0.04
***RANTES (CCL5)***	0.73 ± 0.03	0.73 ± 0.03	0.72 ± 0.02	0.53 ± 0.04
***TNF-α***	0.77 ± 0.03	0.68 ± 0.03	0.80 ± 0.02	0.62 ± 0.04
***VEGF***	0.52 ± 0.03	0.63 ± 0.04	0.52 ± 0.03	0.63 ± 0.04

Bold: highlight the result; Italics: Genes.

**Table 10 cancers-12-03680-t010:** SVM method applied to the tissue-expression dataset. Sex and age were not considered as predictors for the lack of data in the dataset. The *p*, i.e., the probability that the “Accuracy” value is not significantly higher than the “No Info Rate” value, is lower than 0.00001. Hence, we can safely reject the null hypothesis and we can assume that the accuracy of the predictive model is higher than the value of the No Info Rate (0.61), corresponding to the performance of a dummy, fixed-answer predictor.

Num. Melanoma	Num. Controls	Training Set Size	Testing Set Size	AUC (ROC)	Accuracy	No Info Rate	*p*-Value
310	201	358	153	0.99	0.95	0.61	<0.00001

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
