# Peer review of "Investigating Serum and Tissue Expression Identified a Cytokine/Chemokine Signature as a Highly Effective Melanoma Marker"

_cancers, 2020, doi:10.3390/cancers12123680_

Round 1
Reviewer 1 Report
In the study by Cesati et al., authors invetstigated the expression of 27 cytokines/chemokines in serum and tissue samples from melanoma patients and controls. Results obtained showed interesting results, highlighting the potential for a 4-gene signature useful to discriminate melanoma from controls.
I report thefollowing issue to be addressed.
MAJOR POINTS.
Among explored molecules, no matching was observed between differentially expressed genes (melanoma vs control) in serum and tissue samples. How could authors explain/interpret this lack of correspondance?
The identified 4-gene signature, well discriminate melanoma samples from controls, by analyzing tissue samples. However, this kind of evaluation is currently entrusted to pathology and immunohistochemical markers. A diagnostic signature (identified by ROC analysis and AUC value) should be more useful and significant if identified in serum, in order to be adopted for early and non-invasive detection. Please, discuss.
Based on data reported on Table 1, there is a difference between mean age of controls and melanoma patients. Did author perform the statistical analyses to check this difference? In case of p<0.05, how could authors be sure that differences detected between patients and controls are really dependant on disease instead of age? In case of p<0.05, these cohort of samples are not homogeneous.
MINOR POINTS.
In tables, many genes are reported missing part of their names (i.e. MCP-, INF-, TNF-, etc...). Please, integrate.
Author Response
We thank the Reviewer for the useful comments. We carefully addressed all issues raised. Please find below the point-by-point reply.
Reviewer 1:
In the study by Cesati et al., authors investigated the expression of 27 cytokines/chemokines in serum and tissue samples from melanoma patients and controls. Results obtained showed interesting results, highlighting the potential for a 4-gene signature useful to discriminate melanoma from controls.
MAJOR POINTS.
-1) Among explored molecules, no matching was observed between differentially expressed genes (melanoma vs control) in serum and tissue samples. How could authors explain/interpret this lack of correspondance?
Answer:
We thank the Reviewer for this comment, addressing a key point of the manuscript. We did not expect that the same cytokines/chemokines would be modified in serum and in tissues. In fact, in general terms, molecules measured in the serum are likely produced as a systemic response, while molecules measured within the biopsies are directly produced by the tumor or in the regions immediately close to it. Therefore, cytokines/chemokines measured within the biopsies reflect more directly the tumor biology and its aggressive behavior. On the contrary, cytokines/chemokines measured in the serum reflect more directly how the organism responds to the tumor from the inflammatory/immunological point of view. We cannot exclude that molecules produced within the primary tumor may reach the blood. However, such signals may be not measured due to the large dilution in the bloodstream and their expression levels may fall below the detection limit. We used the xMAP technology for quantification in serum samples, to minimize as much as possible sensitivity limitations.
This sentence was introduced in the Discussion of the revised version, to better clarify this point, at row 390-401 of page 23.
-2) The identified 4-gene signature, well discriminate melanoma samples from controls, by analyzing tissue samples. However, this kind of evaluation is currently entrusted to pathology and immunohistochemical markers. A diagnostic signature (identified by ROC analysis and AUC value) should be more useful and significant if identified in serum, in order to be adopted for early and non-invasive detection. Please, discuss.
Answer
We thank the Reviewer for this comment and fully agree. We analyzed serum from melanoma patients with the aim of identifying markers useful for the early diagnosis, using a minimally invasive technique. Expressions significantly different were identified in the sera of melanoma patients vs. controls. However, we could not identify good markers within the 27 molecules investigated. This may depend on the molecules chosen (i.e., we should probably change targets and focus on other molecules), or it may depend on the high dilution factor in the serum or on the variable concentration in the serum.
We introduced a few sentences in the Discussion section to better clarify this issue (row 456-461 of page 25).
We should underline here, that the identification of a 4-genes signature may be a relevant help for pathologists. Measuring the expression of these genes represents a quantitative approach that is operator-independent and may be part of an automatic process useful to identify suspect samples.
We introduced this sentence in the Discussion section to better clarify this issue in row 508-511 of page 26 of the revised version.
-3) Based on data reported in Table 1, there is a difference between the mean age of controls and melanoma patients. Did author perform the statistical analyses to check this difference? In case of p<0.05, how could authors be sure that differences detected between patients and controls are really dependent on disease instead of age? In case of p<0.05, these cohort of samples are not homogeneous.
Answer
Thanks to the Reviewer for this observation.
Age in healthy- and melanoma-groups is statistically different.
The difference reflects what the reality is, i.e., the fact that cancer patients are generally older than healthy controls since increased age is a specific risk factor for cancer. In the present study, individuals were sequentially enrolled, and controls were individuals with a suspect lesion removed and diagnosed by the pathologist as a not-cancer lesion. Therefore, eliminating age differences from the dataset may eliminate a specific cancer risk factor. To have similar age distribution in patients and in controls groups, one would be forced to remove several young healthy controls from the dataset (since young melanoma patients are rare) and to remove several old melanoma patients (since old healthy controls are lacking). This procedure would strongly decrease the number of individuals analyzed, also reducing the statistical potency of the study, and would alter the actual patients- and controls- age distribution. In other words, despite the large number of patients enrolled in the present study (i.e., 232), the age-match in groups already matched for sex, would make groups too small to reach convincing statistical significance. The solution would be to enroll almost twice the number of individuals. We are recruiting new patients, but it will take several more months to reach the necessary numbers. We, therefore, decided to analyze groups matched for sex but not-matched for age, in order to avoid a strong reduction of numerosity, and also to take into account age as a specific component of the cancer patients.
To comply with the Reviewers’ request, we performed age-matching and compared results reported for Table 7, Figure 3, and Table 2 reported in the manuscript.
Age-matching strongly reduced melanoma samples (96 reduced to 65) and controls (136 reduced to 65) to up to almost 50% reduction.
The SVM analysis (the one reported in Table 7) when carried out on the age-matched groups, gave results similar to the ones obtained from un-matched groups (See Supplementary Table 7 in the revised version).
The matrix analysis reported in Fig. 3 (performed on un-matched groups) when carried out on the age-matched groups gave results identical to the ones obtained from un-matched groups.
Finally, the analysis reported in Table 2 was carried out on age-matched-groups. In this case, the 4 molecules found statistically different in Table 2 show very similar fold-changes no matter if age-match is performed or not performed. However, the p values are increased in age-matched groups, due to the much lower numerosity. Namely:
IL1b:
fold change 0.84; p=0.04 in un-matched;
fold-change 0.89; p=0.09 in age-matched.
IP10:
fold-change 1.69; p 0.04 in age-unmatched;
fold change 1.78; p=0.36 in age-matched.
PDGF-BB:
fold change 1.55; p=0.01 in age- unmatched;
fold change 1.41 and p=0.04 in age-matched.
RANTES:
fold change 1.30; p=0.01 in age-unmatched;
fold change 1.29; p=0.03 in age-matched.
Therefore, for data of Table 2, age-matching does not affect strongly fold change but reduces significance due to the strong reduction of numerosity within the groups.
We then conclude that age-match (while required for a correct statistical analysis) in this case strongly reduces the numerosity within the groups and would require much larger numbers of enrolled individuals to keep high statistical potency. In addition, the results and fold changes obtained on age-matched groups appear very similar or identical to the ones obtained on age-unmatched groups.
We thank the Reviewer to highlight this point and introduced a few sentences in the discussion to address this point, at row 462-478 of page 25.
MINOR POINTS.
In tables, many genes are reported missing part of their names (i.e. MCP-, INF-, TNF-, etc...). Please, integrate.
Answer:
We thank the reviewer. We made all corrections and evidenced such corrections in the highlighted version.
Reviewer 2 Report
In the manuscript entitled “Investigating serum- and tissues-expression identifies a cytokines/chemokines signature as a highly effective melanoma marker” Cesati et al. investigated the expression of 27 molecules in serum and cancer tissues of melanoma patients. The reported data could be of clinical relevance in melanoma patients management, however major adjustments are needed to increase clarity and accuracy of the notions reported.
- The major finding of the study was the identification of a gene signature composed by IL-1Ra, IL-7, MIP-1a, MIP-1b. Nevertheless this signature characterizes melanoma specimens and it is different from the one identified in patient sera (IL-1b, IL-6, IP-10, PDGF-BB, and RANTES). How can the author comment on that? Showing two stories with different endings, about patients sera and tissues, ultimately results confusing.
- The expression levels of cytokines and chemokines should be confirmed by immunohistochemical analysis performed on melanoma specimens.
- In order to associate the expression of these markers to prognosis, data should be correlated with patients survival curves.
- Overall, data appear disconnected and should be discussed more homogeneously.
Author Response
We thank the Reviewer for the useful comments. We carefully addressed all issues raised. Please find below the point-by-point reply.
Reviewer 2
In the manuscript entitled “Investigating serum- and tissues-expression identifies a cytokines/chemokines signature as a highly effective melanoma marker” Cesati et al. investigated the expression of 27 molecules in serum and cancer tissues of melanoma patients. The reported data could be of clinical relevance in melanoma patients management, however major adjustments are needed to increase clarity and accuracy of the notions reported.
-1) The major finding of the study was the identification of a gene signature composed by IL-1Ra, IL-7, MIP-1a, MIP-1b. Nevertheless this signature characterizes melanoma specimens and it is different from the one identified in patient sera (IL-1b, IL-6, IP-10, PDGF-BB, and RANTES). How can the author comment on that?
Answer:
We thank the Reviewer for this comment, which addresses a key point of the manuscript. We did not expect that the same cytokines/chemokines would be modified in serum and in tissues. In fact, molecules measured in the serum are likely produced as a systemic response, while molecules measured within the biopsies are directly produced by the tumor or in the regions immediately close to it. Therefore, cytokines/chemokines measured within the biopsies reflect more directly the tumor biology and its aggressive behavior. On the contrary, cytokines/chemokines measured in the serum reflect more directly how the organism responds to the tumor from the inflammatory/immunological point of view. We cannot exclude that molecules produced within the primary tumor may reach the blood. However, they are largely diluted in the bloodstream and their expression levels may fall below the detection limit.
This sentence was introduced in the Discussion of the revised version, to better clarify this point, at row 390-401 of page 23,
-2) Showing two stories with different endings, about patients sera and tissues, ultimately results confusing.
Answer:
We thank the Reviewer for this comment. We revised the discussion to clarify as much as possible the different approaches. We hope clarifications introduced at row 390-401 of page 23, row 462-478 of page 25, and row 508-511 of page 26 improve the clarity.
-3) The expression levels of cytokines and chemokines should be confirmed by immunohistochemical analysis performed on melanoma specimens.
Answer:
We thank the Reviewer for this suggestion. Collecting these data will take some time since new patients should be recruited to this aim and we are requested to have specific authorization from the Ethic Committee. We gladly accept the suggestion and will introduce the immunohistochemical analysis in the results of the next coming manuscript aimed at providing experimental validation of the biopsies expression data. Data of the present manuscript were derived from molecular analyses on 511 patients from GENT2 database and were validated on 1019 patients from GEPIA2 database.
- 4) In order to associate the expression of these markers to prognosis, data should be correlated with patients survival curves.
Answer:
We thank the Reviewer for this suggestion. Studies on five-years survival follow-up on patients recruited in our hospital are currently ongoing and will require 3 more years, since the last patients were recruited in April 2018.
However, to comply with the Reviewer's request, the role of the 4 genes (IL1Ra, IL7, MIP1a, and MIP1b) as survival-prognostic factors was investigated in GEPIA2 database. Interestingly, 3 out of 4 (namely, IL7, MIP1a, and MIP1b) show significant Hazard Ratios, indicating improved survival for high expression values.
We thank the Reviewer for this notice and these results were introduced in the revised version, at row 349-353 of page 21.
-5) Overall, data appear disconnected and should be discussed more homogeneously.
Answer:
We re-phrased and introduced some sentences in the Discussion to make the flow more homogeneous. To improve clarity as much as possible, we kept separate the discussion on serum data from the discussion on tissue expression data. We hope the revised version has improved clarity.
Reviewer 3 Report
Cesati et al. investigated cytokines/chemokines signature profile using melanoma tissue samples and serum of the patients and found that the four gene signature of IL-1Ra, IL-7, MIP-1a and MIP-1b could be biomarkers of melanoma. Overall the paper is well written and interesting. I would like to raise the following minor concerns.
1, There is poor information about the melanoma (primary or metastasis, tumor thickness>1mm or not, and no other information eg. tumor stage). Localized melanomas are unlikely to affect the cytochemokine levels of serum.
2, It will be much more interesting that the four signatures could detect occult micrometastasis.
3, Too many tables and figures. Try to reduce them by half.
Author Response
We thank the Reviewer for the useful comments. We carefully addressed all issues raised. Please find below the point-by-point reply.
Reviewer 3
Cesati et al. investigated cytokines/chemokines signature profile using melanoma tissue samples and serum of the patients and found that the four gene signature of IL-1Ra, IL-7, MIP-1a and MIP-1b could be biomarkers of melanoma. Overall the paper is well written and interesting. I would like to raise the following minor concerns.
- 1) There is poor information about the melanoma (primary or metastasis, tumor thickness >1mm or not, and no other information eg. tumor stage). Localized melanomas are unlikely to affect the cytochemokine levels of serum.
Answer:
We thank the Reviewer for this comment. The serum dataset reports tumor thickness values in all patients; the corresponding data are reported as average in Table 1 and are analyzed in Tables 3 and 4. Unfortunately, tumor thickness and tumor stage values are lacking in many cases regarding biopsy expression; in this case, no statistical analysis is possible regarding Breslow thickness or tumor stage, due to the lack of data. Unfortunately, data from the GENT2 database are not fully annotated, as one may expect.
Regarding the localized melanomas, we agree with the Reviewer that local signals should be probably measured only locally. However, we tested the hypothesis that a small signal generated by the localized melanoma may stimulate a microenvironmental or a systemic response large enough to be detected by a systemic analysis. Using the xMAP technology to measure serum expression reduced as much as possible detection limitations.
-2) It will be much more interesting that the four signatures could detect occult micrometastasis.
Answer:
We fully agree with the Reviewer and thank for this comment since occult micro-metastases represent a serious clinical issue. We accept the suggestion and we are currently collecting histological sections with the aim to correlate gene expression and micro-metastasis occurrence. The study requires the screening of a large patients cohort and will take several more months to reach some relevance.
-3) Too many tables and figures. Try to reduce them by half.
Answer:
We thank the Reviewer. We are aware that this study presents many data and we tried to combine, as much as possible, completeness with clarity. Many Tables are reported as Supplementary materials in order to keep the flow as smooth as possible; one Table has been now inserted as Supplementary material (namely, Table 7 Supplementary). Tables still present in the main manuscript are necessary to fully support the several statistical findings reported and we hope they may represent a useful consultation
Round 2
Reviewer 2 Report
In the revised version of the manuscript “Investigating serum- and tissues-expression identifies a cytokines/chemokines signature as a highly effective melanoma marker” Cesati et al. replied to the reviewer concerns. In particular, the new manuscript version has been improved by better clarification of the role of the different signatures identified in patients’ sera and tissues and the assessment of role of IL1Ra, IL7, MIP1a, and MIP1b as prognostic factors in GEPIA2 database.